# Cell Sources for Cartilage Repair—Biological and Clinical Perspective

**DOI:** 10.3390/cells10092496

**Published:** 2021-09-21

**Authors:** Inga Urlić, Alan Ivković

**Affiliations:** 1Department of Biology, Faculty of Science, University of Zagreb, 10000 Zagreb, Croatia; 2Department of Orthopaedic Surgery, University Hospital Sveti Duh, 10000 Zagreb, Croatia; 3School of Medicine, University of Zagreb, 10000 Zagreb, Croatia; 4Department of Clinical Medicine, University of Applied Health Sciences, 10000 Zagreb, Croatia

**Keywords:** cartilage repair, chondrocytes, stem cells, articular cartilage, autologous chondrocyte transplantation, regenerative medicine, tissue engineering

## Abstract

Cell-based therapy represents a promising treatment strategy for cartilage defects. Alone or in combination with scaffolds/biological signals, these strategies open many new avenues for cartilage tissue engineering. However, the choice of the optimal cell source is not that straightforward. Currently, various types of differentiated cells (articular and nasal chondrocytes) and stem cells (mesenchymal stem cells, induced pluripotent stem cells) are being researched to objectively assess their merits and disadvantages with respect to the ability to repair damaged articular cartilage. In this paper, we focus on the different cell types used in cartilage treatment, first from a biological scientist’s perspective and then from a clinician’s standpoint. We compare and analyze the advantages and disadvantages of these cell types and offer a potential outlook for future research and clinical application.

## 1. Introduction

Cartilage is smooth and elastic connective tissue with four major functions: gives shape and support, cushions joints, allows smooth articulation of bones around each other, and enables the growth of long bones. All these functions make cartilage essential for the musculoskeletal system and movement. Cartilage injuries cause long-term problems among the younger population and athletes, as well as cartilage degeneration and osteoarthritis among the aging population [1]. Regenerative medicine and tissue engineering seek to find solutions for better treatment strategies. The aim of this review is to describe different types of cells that can participate in cartilage regeneration as they are one of the three essential components to be considered for new treatment options (Figure 1).

## 2. Biological Perspective

Cartilage is produced by the chondrocytes, and that makes them the key cells for the development of cell-based cartilage repair and regeneration strategies. Often, these procedures require culture expansion of cells with chondrogenic potential. That property can be attributed to several cell types from the categories of differentiated cells and their unipotent progenitors, as well as multipotent and pluripotent stem cells (Table 1). Among differentiated cells, the well-known sources of cells with chondrogenic capacity are mesoderm-derived articular chondrocytes (MDCs) and neural crest-derived nasal chondrocytes (NCDCs). Less differentiated cells, chondroprogenitors (CPs), are descendants of stem cells but with lineage bias toward chondrocytes and can be isolated from various tissue sources such as cartilage, synovium, meniscus, and infrapatellar pad. Multipotent stem cells with chondrogenic capacity are mesenchymal stem cells (MSCs) from different sources such as bone marrow, adipose tissue, synovia, umbilical cord blood, and peripheral blood. Pluripotent stem cells have even higher differentiation potential and can give rise to all embryonic tissues, including cartilage. The source of pluripotent embryonic stem cells (ESCs) is inner cell mass from the blastocyst stage of embryo development, while induced pluripotent cells (iPSCs) can be derived from differentiated cells using reprogramming techniques.

Besides having an adequate cells source, it is important to induce differentiation to chondrocytes and the production of cartilage extracellular matrix. Chondrocyte differentiation is preceded by condensation of mesenchymal progenitor cells and nodule formation and is regulated by several critical signaling pathways. Signaling molecules that can induce these pathways are bone morphogenetic proteins, fibroblast growth factors, transforming growth factor beta, Wnt, and cell adhesion molecules N-CAM, N-cadherin, and β-catenin) [2]. The master transcription factor is Sox9, and it is essential for the initiation and maintenance of chondrogenesis. By analysis of chondrogenic phenotype, it is possible to assess the quality of chondrocytes and newly formed cartilage. Commonly tested genes or proteins are type II collagen, Sox9, aggrecan, and high levels of proteoglycans. Hypertrophic chondrocytes and fibrocartilage are unwanted destinies of differentiating chondrocytes, and quality testing should confirm the absence or low expression of hypertrophic marker collagen type X and fibrotic marker collagen type I [3].

The third component of the essential trinity is appropriate biocompatible scaffold, mechanical support for cells providing a three-dimensional environment. The most interesting scaffolds for cartilage treatment are resorbable scaffolds whose main function is to provide temporary templates for cells. The cells will attach to the scaffold, proliferate and secrete extracellular matrix as the scaffold resorbs. Eventually, the cartilage matrix will completely replace the scaffold [4]. Scaffold can also be used to deliver bioactive molecules to the damaged tissue areas to guide the growth of the new tissue [5]. Scaffolds can be grouped as natural and synthetic polymers. Natural polymers used for cartilage repair are chitosan, agarose, collagen, hyaluronan, fibrin, and alginate, while common synthetic polymers are alpha-hydroxy esters such as polylactic acid (PLA), polyglycolic acid (PGA), and their copolymer polylactic-co-glycolic acid (PLGA) [5].

Another advancement in the field of cartilage repair and regeneration is the use of chondrons: chondrocytes with pericellular matrix (PCM) [6]. The PCM is a thin layer of extracellular matrix (ECM) that surrounds chondrocytes and supports their function [7]. The use of chondrons provides bioactive mechanical support for cells (with or without scaffold). This can be achieved through increased tissue surface by mincing cartilage tissue into smaller fragments that produce mitogenic signals and activate the migration of chondrocytes and ECM deposition [8].

### 2.1. Differentiated Cells

#### 2.1.1. Articular Chondrocytes (ACs)

Chondrocytes are the resident cell type in the articular cartilage that secrete extracellular matrix and sustain the cartilage. They can easily be isolated from intraoperatively harvested cartilage tissue, and even from post-mortem and refrigerated tissue [9], simply by dissection followed by enzymatic digestion. Articular chondrocytes (ACs) have been used for cartilage tissue engineering due to their ability to proliferate in monolayer culture. However, their expansion in monolayer cell culture leads to the gradual loss of their differentiated phenotype. The process of dedifferentiation can be followed using specific markers such as the loss of collagen type II expression and increase in collagen type I expression. After only two passages in culture, there is a ten-fold decrease in the ratio of type II/type I collagen that decreases further rapidly with expansion [10,11]. If extensive expansion of ACs in culture limits their redifferentiation potential, that could affect neocartilage formation after ACs implantation [12]. Different studies have confirmed the latter findings, concluding that chondrocytes lose their ability to redifferentiate after approximately 5–6 population doublings [13,14,15]. However, the redifferentiation potential can be restored upon transfer into a three-dimensional culture system and medium supplementation with specific regulatory molecules [16]. Re-expression of type II collagen and glycoprotein aggrecan during the expansion of human adult chondrocytes through 12 population doublings was achieved by adjusting the growth conditions using serum-free medium with the presence of transforming growth factor beta 1 or 2 (TGF-β1 or 2) and insulin-like growth factor 1 (IGF-1) or insulin [17]. Therefore, the serum-free medium formulations containing TGF-β family members, IGF-1, fibroblast growth factor 2 (FGF-2), and platelet-derived growth factor BB (PDGF-BB) are referred to as standard chondrogenic medium [9,16].

#### 2.1.2. Nasal Septum Chondrocytes (NCs)

Nasal septum cartilage, such as articular cartilage, is hyaline cartilage made of extracellular matrix produced by nasal septum chondrocytes (NCs). These chondrocytes have different origins from articular chondrocytes as they migrated from the neural crest during embryonic development. It has been shown that they have an even greater ability to repair articular cartilage than articular chondrocytes [18,19]. NCs can be isolated from nasoseptal cartilage biopsy performed under local anesthesia followed by dissection and enzymatic digestion [18]. NCs can be expanded in monolayer culture with high proliferative capacity (Figure 2) [20]. The phenotype changes during monolayer culture are similar to those happening to ACs as the expression of cartilage-specific genes such as collagen type II decreases with serial expansion [21]. However, NCs have greater potential to redifferentiate to chondrocytes and form hyaline cartilage in a three-dimensional culture system. The addition of chondrogenic medium containing TGF-β, IGF-1, or other supplements can further enhance the expression of cartilage markers [22,23].

Besides easy harvesting, high proliferation capacity, and cartilage-forming potential, NCs have other advantages over ACs. They are less dependent on the donor age [20], and cartilage formation is more reproducible [24]. They also exhibit features of self-renewal capacity and HOX-negative expression profile that can be reversed after implantation, suggesting their high plasticity [25]. All these properties of NCs make them promising candidates for craniofacial and articular cartilage regeneration.

### 2.2. Progenitors

#### Chondroprogenitors (CPs)

Chondoprogenitors (CPs) refer to a population of progenitor cells that are specifically predisposed to differentiate into chondrocytes. CPs have been isolated from various tissues, including cartilage, synovium, adipose tissue, meniscus [26,27]. Chondroprogenitor population can be isolated and purified from those tissues after digestion with type I or II collagenases followed by cell sorting for specific MSC-associated cell surface markers CD105, CD9, CD90, CD166, and CD146 [28]. The advantages of CPs for clinical applications are that they are already primed for chondrogenic differentiation, and expansion does not alter differentiation, but their disadvantage is limited abundance due to the use of autologous sources [26]. Extensive expansion in vitro is necessary for CPs clinical use, and it has been successful both in standard conditions and in the presence of high fetal calf serum (FCS) concentrations up to 40% and in the presence of TGF-β1 [29]. However, scalability still remains the problem for clinical use.

### 2.3. Mesenchymal Stem Cells (MSCs)

Mesenchymal stem cells (MSCs) are multipotent stem cells with the ability to differentiate into specialized cells developing from mesoderm. They are present in multiple tissues, but the well-recognized sources are bone marrow, adipose tissue, and umbilical cord blood. International Society for Cellular Therapy (ISCT) defined criteria that cultured MSCs have to meet, and those are: adherence to plastic dishes for cell culture; positivity for markers CD73, CD90 and CD105, and absence of CD45, CD31, CD34, CD14, CD11b, CD79alpha, CD19, and HLA-DR; and can differentiate into osteoblasts, chondrocytes, and adipocytes under appropriate conditions [30]. MSCs have various clinical applications based on their anti-inflammatory and immuno-modulatory properties, their ability to repair endogenous tissues as well as the ability to differentiate in different cells of mesenchymal origin [31]. Their chondrogenic potential enabled the development of regenerative medicine and tissue engineering approaches to treat cartilage defects. In the text below, significant sources of MSCs have been described together with their advantages and disadvantages.

#### 2.3.1. Bone Marrow-Derived Stem Cells (BM-MSCs)

The most common adult source of MSCs for cartilage tissue engineering is bone marrow. It is harvestable and renewable tissue that can be used in an autologous or allogenic manner. The collection of bone marrow is usually achieved by aspiration from the iliac crest of the pelvis. From 25 mL of bone marrow aspirate, 100–150 million bone marrow-derived stem cells (BM-MSCs) can be produced in 3 weeks, giving the volume of 0.4–0.5 mL [32]. They can be successfully expanded to generate a sufficient number of cells while retaining their ability to differentiate [9]. For example, the addition of FGF-2 supplementation prolongs the life span of BM-MSCs to more than 70 doublings and maintains their differentiation potential until 50 doublings [33]. However, there is a certain variability in the number of isolated cells, clonogenicity, and differentiation potential in the population as the fitness of BM-MSCs decreases with aging [34]. In spite of these problems, the use of BM-MSCs for cartilage repair relieves the problems of chondrocyte harvesting, donor site morbidity, and all the limits of chondrocyte growth in culture. BM-MSCs, under appropriate culture conditions, can be driven down the chondrocyte lineage. The most commonly used method involves culturing of BM-MSCs in chondrogenic medium as cell aggregates referred to as pellet culture [9]. Pellet culture is in vitro model resembling mesenchymal precartilage condensations in embryos. It is favorable for cartilage formation because it mimics cell-cell interactions. However, bioactive signals such as dexamethasone and TGF-β1 are an important addition to growth medium [35]. Approximately after 14 days in chondrogenic medium, but dependent on donor, hyaline-like tissue is formed by the differentiating cells. The new modified chondrogenic medium composition was developed by Mackay et al. [36], and it contains TGF-β, dexamethasone, ascorbic acid-2-phosphate, proline, and ITS premix (insulin, transferrin, selenium, linoleic acid). The major challenge of chondrogenic induction of BM-MSCs is controlling their differentiation because BM-MSCs tend to exhibit a hypertrophic phenotype leading to calcification [37,38]. The chondrogenic differentiation can be improved by coculture of BM-MSCs with chondrocytes. It has been suggested that coculture produces a better cartilage matrix due to the BM-MSCs’ trophic role rather than their active chondrogenic differentiation [38,39,40].

#### 2.3.2. Adipose Tissue-Derived Stem Cells (AD-MSCs)

The second most common source of tissue for MSCs is the adipose tissue and adipose tissue stromal vascular fraction (SVF) [41]. Adipose tissue is also harvestable tissue, and unlike bone marrow, it is considered unwanted. So, the great advantage of adipose tissue-derived stem cells (AD-MSCs) over BM-MSCs is their abundance and accessibility, as well as their clonogenic potential [42]. Even though AD-MSCs differentiate from chondrocytes, they were shown to possess lower chondrogenic potential and produce less matrix than their bone marrow-derived counterparts [43,44].

The adipose tissue is obtained using a liposuction procedure, and it is referred to as the lipoaspirate. After digestion with collagenase followed by rinsing, the cell mixture is called stromal vascular fraction (SVF) and contains AD-MSCs, fibroblasts, white and red blood cells, endothelial cells, etc. The AD-MSCs from SVF can be further isolated and expanded [45]. Various protocols have been established to achieve chondrogenic differentiation of AD-MSCs [46], and one of the examples of chondrogenic medium contains Dulbecco’s Modified Eagle Medium (DMEM) TGF-β3, albumin, dexamethasone, ascorbic acid, transferrin, and insulin [47]. In addition, oxygen deprivation (1%) enhances AD-MSCs proliferation and 5% promotes chondrogenesis [48]. AD-MSCs are adherent cells that prefer growth in monolayer culture. However, the accumulation of cells in pellet culture is necessary for chondrocyte differentiation. This led to the development of three-dimensional scaffolds that would overcome the growth inhibition and support chondrogenesis [49]. The application of mechanical forces to cells during culture in vitro also improves chondrogenesis. Even though the molecular mechanisms have not been understood yet, it is speculated that cellular actin filaments were key initial regulators of cell morphology in response to mechanical forces from the extracellular environment [50].

#### 2.3.3. Synovium-Derived Stem Cells (Sy-MSCs)

The synovium is a thin membrane that lines the cavity of synovial joints. It produces the synovial fluid for joint lubrication and cartilage nutrition. For the first time, synovium-derived stem cells (Sy-MSCs) were isolated from the synovium of human knee joints in 2001. by De Bari et al. [51]. The source of Sy-MSCs is usually discarded fragments at arthroscopic surgery or infrapatellar fat pad and synovial fluid obtained by the minimally invasive and routine arthroscopic procedure [52,53]. Sy-MSCs are characterized by the expression of surface markers CD44, CD105, CD73, CD166, CD90, CD106, STRO-1, and by low or no expression of CD34, CD45, CD14, and HLA-DR [54]. CD44 is the hyaluronic acid receptor, and its expression is considered to be a true property of Sy-MSCs [55]. It has been shown that Sy-MSCs had a greater proliferation capacity and stronger chondrogenic potential than BM-MSCs and AD-MSCs, as well as less hypertrophic differentiation than bone BM-MSCs [56]. In addition, pellet cultures were significantly larger from Sy-MSC than BM-MSCs in patient-matched comparisons [57]. Proliferation capacity is maintained even after 10 passages of Sy-MSCs, and it does not depend on donor age or location of their collection. Differentiation ability into chondrogenic lineage was also superior when compared to other sources of MSCs, regardless of donor age or location of their collection [51,54]. All these properties, together with the same embryonic origin as articular cartilage, make these cells a suitable choice for the treatment of cartilage defects. A combination of bone morphogenetic protein (BMP-2), TGF-β, and dexamethasone was optimal for the induction of chondrogenesis Sy-MSCs [57]. In addition, the research of Roelofs et al. showed that Sy-MSCs carry an imprinted code for joint morphogenesis. They injected adult human Sy-MSCs overexpressing bone morphogenetic protein 7 (BMP-7) in the skeletal muscle and observed an ectopic formation of joint-like structure, providing evidence of their morphogenetic properties [58].

#### 2.3.4. Umbilical Cord Blood-Derived Mesenchymal Stem Cells (UC-MSCs)

Umbilical cord blood is a promising new source of allogenic MSCs due to easy non-invasive harvesting. In addition, umbilical cord blood-derived mesenchymal stem cells (UC-MSCs) became commercially available. UC-MSCs are abundant in collected cord blood samples, and due to their low immunogenicity, they do not cause an immune response post-transplantation [59,60]. They have been compared to MSCs from other sources (BM-MSCs and AD-BMCs), and their isolation success rate was significantly lower, but their proliferation capacity was the highest. The expression of surface markers is typical for MSCs, but the intensity of CD90 and CD105 expression is lower than what is detected in BM-MSCs and AD-BMCs [61]. They do possess intrinsic chondrogenic potential, but the protocols for their chondrogenic differentiation are still being developed. Gomez-Leduc et al. demonstrated that UCB-MSCs could be the choice for the treatment of cartilage defects when they are seeded on the collagen sponge scaffold in the presence of BMP-2 and TGF-β1 under normoxic conditions [62].

#### 2.3.5. Peripheral Blood-Derived Mesenchymal Stem Cells (PB-MSCs)

Circulating MSCs are also called peripheral blood-derived mesenchymal stem cells (PB-MSCs). The frequency of PB-MSCs in humans is very low, in the order of 1 in 10^8^ peripheral blood mononuclear cells (MNCs) [63]. For comparison, the frequency of BM-MSCs in bone marrow is 1 in 10^4^ to 1 in 10^5^ bone marrow MNCs [64]. The BM-MSCs and PB-MSCs showed similar characteristics of cell proliferation and multi-differentiation potentials. Xu and Li compared the expression of PB-MSCs and BM-MSCs surface markers and found that CD73 may be an important indicator to distinguish them. PB-MSCs are also plastic-adherent and have multi-differentiation potential, fulfilling the criteria of the International Society for Cellular Therapy [65]. Peripheral blood became an alternative source of MSCs as they can be harvested easily with minimally invasive procedures and isolated by density gradient centrifugation. PB-MSCs can be used in an autologous and allogenic manner, and also, they can be used as a non-cultured or culture-expanded treatment for cartilage repair and regeneration [66]. Some researchers argue that the number of PB-MSCs in circulation is low [67], but their number can be increased in the bloodstream by stimulation using granulocyte colony-stimulating factor (G-CSF). The technique is called “blood mobilization” and results in the production of mixed cells: MSCs, immature progenitor cells, and hematopoietic stem cells [68]. PB-MSCs need to be isolated, expanded, and grown in chondrogenic medium to ensure differentiation toward the cartilage lineage. Even though the number of isolated BMSCs is higher, 5 million PB-MSCs can be expanded in vitro just from 2 mL of peripheral blood, exceeding the number of cells necessary for many cartilage repair procedures. Differentiation potential to chondrocytes is similar to that of BM-MSCs [69].

### 2.4. Pluripotent Stem Cells

MSCs are multipotent cells and can differentiate into several cell types of common origin. On the other hand, pluripotent stem cells can differentiate into all cell types of embryonic origin, excluding only extraembryonic tissues. Pluripotent stem cells can be derived using several techniques, and here, we will focus on embryonic stem cells (ESCs) derived from inner cell mass from the blastocyst stage of the embryo and induced pluripotent stem cells (iPSCs) derived by reprogramming of adult somatic cells.

#### 2.4.1. Embryonic Stem Cells (ESCs)

During embryonic development, chondrocytes are derived from cells of mesenchymal origin [70]. Therefore, the choice of adult somatic cells for cartilage regeneration are multipotent MSCs derived from different sources. Since there are some limits in the regeneration potential of these cells, especially when looking at the cartilage tissue quality, other approaches are considered. If we take the step back looking at the cartilage development, all MSCs are derived from the inner cell mass of the blastocyst. That makes ESCs an attractive allogenic alternative due to their unlimited number and high developmental plasticity. ESCs have been employed to obtain mesenchymal progenitors and chondrocytes [71,72,73]. The methodology includes the generation of embryonic bodies, sorting of mesenchymal cells using surface markers CD105 or CD73, and further culturing on murine cell lines or coculturing with primary chondrocytes [73,74]. However, there are certain problems with clinical applications of ESCs as the efficient protocols for their differentiation into functioning chondrocytes as well as the safety risk of residual undifferentiated cells that can have a tumorigenic potential.

#### 2.4.2. Induced Pluripotent Stem Cells (iPSCs)

Takahashi and Yamanaka, in 2007, successfully reprogrammed human adult somatic cells to an undifferentiated pluripotent state resembling ESCs [75]. The reprograming of fibroblasts was achieved by retroviral delivery of four transcription factors (*Oct4, Sox2, c-Myc,* and *Klf4*) responsible for activation of self-renewal and maintenance of the pluripotent state. Since then, the reprogramming of a variety of cell types has been achieved using different combinations of genes and proteins that are involved in self-renewal and pluripotency. Cells obtained using reprogramming techniques are named induced pluripotent stem cells (iPSCs). Even though ESCs and iPSCs share similar characteristics regarding morphology, surface marker expression, and gene expression profiles [76], iPSCs are superior for clinical applications. They are easily accessible, autologous, and can bypass ethical issues because they are not in the focus of political and religious views such as ESCs [77]. The pluripotency of iPSCs makes them a suitable candidate for applications in cartilage regeneration. That was confirmed by spontaneous generation of cartilage in iPSCs-caused teratomas [75]. In addition, there are several established protocols for chondrogenic differentiation of iPSCs, and they include coculture with primary chondrocytes or use of conditioned medium from cultured chondrocytes, differentiation through embryoid body formation, differentiation induced by a combination of growth factors, and differentiation from the intermediate population of MSC-like cells [78,79]. It is difficult to draw the conclusion that of all the developed methods produces the best chondrocytes because protocols use different iPSC lines derived from different somatic cells and are reprogrammed using different methods [77]. The same is with protocols for cartilage tissue engineering where iPSC-derived chondrocytes grow in three-dimensional cultures or scaffolds to form cartilage-like tissue in vivo or in vitro [78]. Therefore, robust, reproducible protocols inducing uniform differentiation for clinical applications are still not available. As with other pluripotent cells, safety considerations need to be taken into account due to the possible re-emergence of undifferentiated cells that can represent a possible risk for tumorigenesis [80].

## 3. Clinical Perspective

### 3.1. Differentiated Cells

#### 3.1.1. Articular Chondrocytes (ACs)

In the early 1970s, Bentley and Greer [81] published the paper in Nature, where they described transplantation of the isolated epiphyseal and articular cartilage chondrocytes into joint surfaces of rabbits. Followed by this pioneering work and after another two decades of research, Brittberg et al. developed cell-based therapy to treat articular cartilage defects and reported early clinical results after treating 23 patients with knee cartilage defects [82]. Autologous chondrocyte implantation (ACI) marked the beginning of a tissue engineering era in orthopedic surgery. ACI is basically a three-step process. The first step consists of arthroscopical harvesting of cartilage from the non-weight-bearing portion of the joint. During the second step, chondrocytes are enzymatically released from the specimen and culture-expanded for another 4 to 6 weeks. During the third and final step, the cells are implanted into the defect via (mini)arthrotomy. Original ACI technique injected chondrocyte-suspension and a periosteal flap secured with transcartilagineous sutures and additionally sealed with fibrin glue. In the next iteration, the periosteal cover was replaced with collagen membrane cover, and the final modification included a variety of scaffolds seeded with chondrocytes (known as matrix-assisted chondrocyte implantation or MACI). Although the standard technique requires two procedures and involves arthrotomy, a minimally invasive full-arthroscopic technique has been described as well [83]. A further modification of the ACI strategies is the use of chondrocytes cultured as small spheroids that are totally autologous without the use of any foreign material and are already in a higher developmental state than chodrocytes in suspension [84,85]. Indications for ACI include larger lesions (4–10 cm^2^), younger and more active patients. It should be noted that outcomes tend to be much poorer in patients with previous attempts of cartilage repair such as bone marrow stimulating techniques (microfractures) as well as in patients with a presence of osteoarthritis (OA) [86]. Although long-term results seem to be favorable for an appropriate indication, graft survival rates are suboptimal, being 78% at 5 years and 51% at 10 years [87,88]. To overcome the limitations of ACI, such as the need for cells expansion, high cost, and invasiveness, minced cartilage implantation (MCI) has evolved as an alternative strategy [89]. It is a single-step approach where cartilage fragments are collected during arthroscopy, minced and mixed with platelet-rich plasma (PRP) paste, and applied arthroscopically into the defect [90] (Figure 3). Even less invasive approach has been reported by Marcarelli et al. [91], where they used autologous micrografting technology to deliver chondrocyte-suspension to the joint.

#### 3.1.2. Nasal Chondrocytes (NCs)

NCs emerged as a very interesting alternative for cartilage tissue engineering. From the clinical standpoint, they are much more accessible via simple outpatient procedures, cause much less morbidity, and are much more resilient to inflammatory environments [92]. A preclinical large animal study on goats confirmed the safety and feasibility of the approach [93]. Vukasovic et al. reported on the use of bioreactor-based NC-derived cartilage implants and their application in a sheep model [94]. Collectively, all the available data showed very promising results, which led to the observational first-in-human clinical trial performed by joint efforts of clinicians and scientists from the University Hospital in Basel [19]. In this study, 10 patients with symptomatic, post-traumatic, full-thickness cartilage lesions measuring 2–6 cm^2^ were treated with implantation of NC-engineered grafts. During an outpatient procedure, chondrocytes were harvested from the nasal septum in local anesthesia, expanded, and seeded onto collagen membranes. During the second procedure, mature grafts were transplanted into the knee cartilage defects (Figure 4). Patients were followed for a minimum of 24 months, during which no adverse reactions were recorded, and self-reported clinical scores (Knee injury and Osteoarthritis Outcome Score/KOOS and International Knee Documentation Committee/IKDC) showed significant improvement from the baseline. Based on the promising results obtained in the phase I study, a phase II multicenter, prospective clinical study is ongoing in five European clinical centers (ClinicalTrials.gov Identifier: NCT02673905).

### 3.2. Mesenchymal Stem Cells (MSCs)

#### 3.2.1. Bone Marrow-Derived Stem Cells (BM-MSCs)

BM-MSCs are the most commonly used stem cells for cartilage repair. Intra-articular application of BM-MSCs has been successfully tested in numerous smaller clinical trials to treat osteoarthritis (OA), and it resulted in clinical improvement and pain reduction [95,96,97,98]. There are also reports of cartilage morphology improvement as demonstrated by histological analysis [99]. Focal cartilage defects have also been treated successfully with BM-MSCs. Initial experience with two patients treated for patellar defects was reported by Wakitani et al. [100]. During the decade between 1998 and 2008, the same group reported on favorable results involving 45 joints in 41 patients following the repair of focal cartilage defects with autologous BM-MSCs [101]. Different materials, such as collagen or hyaluronic acid, have been used as carriers for BM-MSCs. Buda et al. [102] used hyaluronic acid membrane as a scaffold for autologous BM-MSCs and treated 20 consecutive patients with osteochondral (OCD) knee lesions using a one-step arthroscopic technique. They found significant improvement in both the IKDC and the KOOS score, as well promising histochemical and immunohistochemical analysis of the biopsy specimens. In another study, 23 patients were followed prospectively for a mean 8 years after treatment of full-thickness cartilage injury with hyaluronic acid-based scaffold embedded with bone marrow aspirate concentrate [103]. Significant improvements in the Tegner and the IKDC score were observed. In a recent publication, Chimutengwende-Gordon [104] reported on a single-step surgical technique for transplantation of BM-MSCs seeded on a collagen sponge to treat OCD of the knee. Nejadnik et al. [105] conducted a study that compared clinical outcomes of patients (*n* = 36) treated with first-generation ACI to patients (*n* = 36) treated with autologous BM-MSCs. Although both methods demonstrated significant improvement in quality of life, health, and sport activity, no difference between the ACI and BM-MSCs groups has been observed. They reported that younger patients (<45 years) in the ACI group had better outcomes, but no difference with respect to the age of patients has been observed in the BM-MSCs group.

#### 3.2.2. Adipose Tissue-Derived Mesenchymal Stem Cells (AD-MSCs)

From the clinical standpoint, easily accessible subcutaneous fat tissue and simple isolation protocols are obvious advantages. Many different clinical studies evaluated the use of AD-MSCs for the treatment of knee OA [45,106]. In most of these studies, AD-MSCs are prepared in the form of a stromal vascular fraction (SVF), which is obtained immediately after collagenase digestion and contains different proportions of AD-MSCs along with the other components such as pericytes, fibroblasts, monocytes, and erythrocytes. In addition, in mentioned studies, authors used fibrin or PRP or both as an addition to AD-MSCs, which further obscures the specific role of each SVF component [107]. In a different approach, Kyriakidis et al. [108] harvested subcutaneous fat, AD-MSCs were expanded and identified (according to the criteria of the International Society for Cellular Therapy), embedded into the hyaluronic scaffold, and then transplanted into the chondral defect. Clinical outcomes demonstrated significant improvements in all subcategories of the KOOS score, the IKDC subjective score, Tegner Activity Score, and VAS score. Jo et al. [22] used AD-MSCs prepared from the abdominal subcutaneous fat by liposuction and expanded in the lab. After standard arthroscopic examination, cells were resuspended in 3 mL of saline and injected into the knee joint. Primary outcomes were the safety and the Western Ontario and McMaster Universities Osteoarthritis Index (WOMAC) at 6 months after injection, and secondary outcomes included four categories: clinical, radiological, arthroscopic, and histological. The results demonstrated that intra-articular injection of AD-MSCs improved function and pain of the knee joint without causing adverse events and repaired cartilage defects by regeneration of hyaline-like articular cartilage. Another approach involves intra-articular instillation of AD-MSCs following the treatment of the focal cartilage lesions with microfractures [109]. In this study, patients were divided into two groups: one group treated with arthroscopic marrow stimulation treatment alone (group A) and the second group who underwent AD-MSCs injection along with arthroscopic marrow stimulation treatment (group B). Clinical outcomes were evaluated according to the visual analog scale (VAS) for pain, the American Orthopedic Foot and Ankle Society (AOFAS) Ankle-Hindfoot Scale, and the Roles and Maudsley score. The Tegner activity scale was used to determine outcomes in activity levels. The authors concluded that injection of MSCs in addition to marrow stimulation treatment had beneficial effects, especially if the lesion size was larger than 109 mm^2^ or a subchondral cyst existed.

#### 3.2.3. Synovium-Derived Mesenchymal Stem Cells (Sy-MSCs)

Sy-MSCs have greater chondrogenic potential than cells from other sources, but on the other hand, are relatively difficult to obtain. Small animal studies demonstrated that Sy-MSCs adapted well to local microenvironments and differentiated into chondrocyte-like cells [110]. Sy-MSCs used in combination with PRP gel resulted in the restoration of osteochondral defect in a rabbit model [111]. The effectiveness of this approach might be limited by the small number of cells, and bigger defects have not been tested. In order to overcome this limitation, Kondo et al. [112] evaluated the possible use of Sy-MSC aggregates to treat osteochondral defects in minipigs. The aggregates of 250,000 Sy-MSCs were formed, and 16 aggregates (for each defect) were transplanted on osteochondral defects measuring 6 × 6 × 1.5 mm in the medial femoral condyle and femoral groove. The knee joints were evaluated at 4 and 12 weeks by macroscopic findings and histology, and the results demonstrated significantly better improvement in the Sy-MSCs group when compared to empty controls (created on the contralateral knee). Promising pre-clinal results encouraged first human clinical trials. In a prospective, randomized, single-blind pilot study, Akgun et al. [113] treated 14 patients and compared those treated with Sy-MSCs embedded in a matrix collagen membrane with those treated with standard MACI. At a 2-year follow-up, both groups revealed significant improvement from the baseline, without any significant differences between the groups. In another case, Sy-MSCs were harvested, cultured in autologous human serum, and arthroscopically implanted in 10 patients [114]. At 3-year follow-up, Lysholm score and MRI appearance improved, but no benefit was reported in Tegner activity level. In addition, the authors proposed three potential advantages of their approach: (1) expanded passage 0 Sy-MSCs were ready in 14 days for transplantation; (2) the cells we transplanted arthroscopically, and (3) scaffolds were not used.

#### 3.2.4. Umbilical Cord Blood-Derived Mesenchymal Stem Cells (UC-MSCs)

Since there are no ethical concerns associated with their use, and there is no donor site morbidity associated with other sources such as bone marrow of subcutaneous fat. Additional advantages include the possibility to use them as off-the-shelf products and the fact that these cells do not require tissue matching for allogenic transplantation [115]. Animal studies demonstrated the feasibility of using UC-MSCs in real-world clinical scenarios. Ha et al. treated osteochondral defect in a minipig model with a mixture (1.5 mL) of human UC-MSCs (0.5 × 10^7^ cells per milliliter) and 4% hyaluronate (HA) hydrogel composite. At 12 weeks postoperatively, the transplanted knees demonstrated superior cartilage restoration in comparison with empty controls from the contralateral knee. In a similar approach, Park et al. [116] treated osteochondral defects created in the rabbit trochlear groove with human UC-MSCs and HA composite. The animals were euthanized 8 and 16 weeks after the index procedure and histological analysis confirmed superior cartilage restoration in the treated knees. In addition, immunohistochemical analysis with anti-human antibodies confirmed the gradual disappearance of the transplanted MSCs. Based on these promising preclinical results, the first human trials followed. Park et al. [117] reported on the first-in-human clinical trial investigating the safety and feasibility of transplantation of a composite made of UC-MSCs and HA hydrogel in seven patients. During a 7-year follow-up, no significant adverse events were observed, and the improved clinical outcomes remained stable. In another study, 93 patients underwent treatment of full-thickness chondral defects in the OA knee with a high tibial osteotomy (HTO) and transplantation of UC-MSCs [118]. The patients were followed for a minimum of one year, and at the final follow-up, the median IKDC subjective score, the WOMAC score, the KSS pain and function scores, and the HSS improved significantly. Yang et al. [119] compared clinical and second-look arthroscopic outcomes between bone marrow aspirate concentrate (BMAC) augmentation and UC-MSCs transplantation in HTO for medial compartmental knee OA. At a mean follow-up of 33 months, clinical outcomes including IKDC, KOOS, SF-36, and Tegner activity scores were significantly improved in both groups, but there were no differences between the two groups. However, second-look arthroscopy showed better healing of repaired cartilage in the UC-MSCs group. Similar results were obtained in a recent retrospective study of 150 cases that underwent HTO with MFX combined with BMAC or allogeneic UC-MSCs procedure for medial unicompartmental OA [120]. A total of 123 cases underwent plate removal and second-look arthroscopy after a minimum of 1 year after the HTO surgery. Morphology of the repaired cartilage was superior in the UC-MSCs group, but there were no differences in the clinical outcomes between the groups.

#### 3.2.5. Peripheral Blood-Derived Mesenchymal Stem Cells (PB-MSCs)

Since they are easily accessible and extractable from the peripheral blood, PB-MSCs represent an attractive alternative as a cell source for cartilage repair strategies. Initial reports of clinical use came from Poland, where investigators presented early results of talus cartilage defects treated with autologous MSCs CD34+ implantation technique [121]. They treated 9 patients and observed improved Magee score as well as magnetic resonance-determined (MR) morphology of the repaired cartilage resembling native one. A few years later, the same group published the results of the study where they compared clinical outcomes following the reconstruction of the knee osteochondral lesions in two groups of patients treated with PB-MSCs and with BMAC [122]. The authors reported a significant improvement across all scales in 86% of all treated patients. They also noted a statistically significant superiority of the group treated with PB-MSCs. At 5 years, a slight decrease in mean clinical assessment scores was seen in both groups of patients. Another approach involves the intra-articular injection of PB-MSCs to enhance chondrogenesis. In a randomized control trial, 50 patients with International Cartilage Repair Society (ICRS) grade 3 and 4 lesions of the knee joint underwent arthroscopic subchondral drilling; 25 patients each were randomized to the control (HA) and the intervention (PB-MSCs + HA) groups [123]. Both groups received five weekly injections commencing 1 week after surgery.

### 3.3. Pluripotent Stem Cells

#### 3.3.1. Embryonic Stem Cells (ESCs)

It has been shown that under optimal conditions, the hESC-derived cells proliferate without phenotypic changes and maintain MSCs surface markers [124]. When put into specific hydrogels, they can generate neocartilage-producing cartilage-specific gene upregulation and extracellular matrix production [125]. From the theoretical standpoint, because of their observed capability of unlimited self-renewal and multilineage differentiation, they could be a potentially unlimited source for cartilage engineering. However, ESCs have shown to be very problematic due to their pluripotency, which is difficult to control, and there is a considerable risk of teratoma formation. In addition, ESCs induce the risk of immune rejection, and there are considerable ethical concerns about the use of human embryos for this purpose. Evens though it is obvious that ESCs at this point have no future in cartilage engineering, recent studies showed promising results in the use of exosomes from ECSs to alleviate OA [126].

#### 3.3.2. Induced Pluripotent Stem Cells (iPSCs)

Since their discovery in 2006, iPSCs have generated tremendous interest in both the scientific and clinical communities [127]. The idea that a very large number of autologous cells can be derived from a small starting population of cells is very appealing and holds great promise for cartilage tissue engineering. iPSCs exhibit similar pluripotency and abundancy as ECSs but without problems associated with ethical and political issues [128]. The current preclinical focus is directed toward developing robust and reproducible chondrogenic differentiation protocols that will allow the production of a uniform chondrocyte population to be used in cartilage repair strategies [129]. At this point, this goal has not been reached, and the direct consequence is the lack of human studies and a relatively small number of animal studies (especially on relevant translational models) (Figure 5) [77]. In one such study, porcine iPSCs were transplanted into an osteochondral replacement model after minimal treatment in vitro, and cartilage regeneration was observed without tumor formation [130].

## 4. Conclusions

Cells are the most important component of regenerative medicine and tissue engineering strategies. As signals are the force of differentiation, and scaffolds are temporary shape and support, the true creators of the new tissue are the cells. Therefore, the selection of cell types and their source is an important topic of the field.

Developmental biology provided us an insight into where to look for cells that have an intrinsic ability to become chondrocytes and secrete cartilage. Our totipotent zygote divides and becomes blastocyst with pluripotent cells that can give rise to all embryonic tissues. Further development restricts differentiation potential to multipotent cells that can give rise to similar cell types. Mesenchymal stem cells are the precursor of chondrocytes and are found in many different tissues, as described above. Further, they prime to chondrogenic lineage as chondroprogenitors. As they become terminally differentiated chondrocytes, they are found embedded in the cartilage matrix that they actively secrete.

Every stage of cell differentiation from pluripotency to fully differentiated cells has advantages and disadvantages in regard to clinical application. The most relevant properties are harvesting, availability, proliferation, chondrogenic potential, ethical and safety considerations as well as the possibility of allogenic or autologous applications (Table 1). Pluripotent cells such as ESCs and iPSCs can have unlimited proliferation potential, but protocols for differentiation and safe use are still developing. Differentiated cells such as ACs have been used for almost three decades in clinical procedures, and ACI represents the golden standard in cell-based cartilage repair at the moment. However, ACI has obvious limitations with a constant need for improvement. One of those improvements is the *nose-to-knee* (N2K) strategy involving the use of NCs to repair articular cartilage. Not only that the harvesting of NCs is much easier with lower patient morbidity, but the cells have much better biological potential and are much more resilient to a postoperative inflammatory environment. The common disadvantages of the use of progenitors and terminally differentiated cells are low availability, difficult harvesting, and donor site morbidity.

Autologous transplants are the golden standard when possible, but it is also important to emphasize some advantages of allogenic sources. Using allogenic cells from different donors enables patients to undergo a single surgical procedure. The tissue source is not as limited, and the risk of infection is lower. Allogenic transplants may carry the risk of immune rejection and disease transmission, so the requirements to produce clinical-grade standardized off-the-shelf products are high.

The highest clinical potential lies in the application of multipotent MSCs. MSCs are not as uniform as differentiated cells, can be harvested from many tissues, and require optimization and adaptation of protocols as well as thorough characterization of newly formed tissue. Accessible sources of MSCs are umbilical cord and peripheral blood, but the abundance of the MSCs is low. The best chondrogenic potential seems to have MSCs from synovium, but the harvesting is difficult and abundance very low. Taken all together, the most favorable results are obtained with bone marrow-derived MSCs with moderate results in all categories.

## Figures and Tables

**Figure 1 cells-10-02496-f001:**
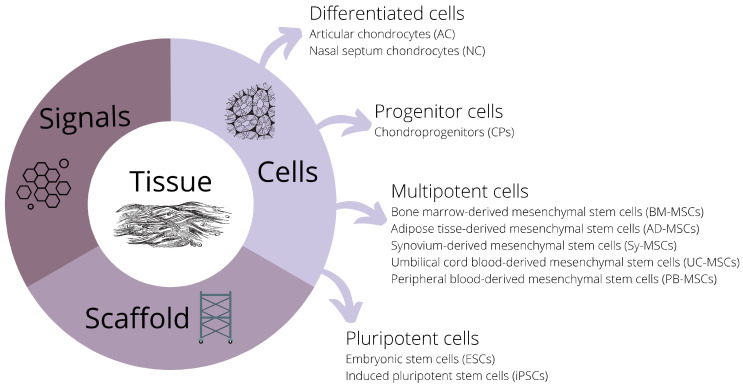
Three essential components of new strategies for cartilage treatment: cells, signals, and scaffolds.

**Figure 2 cells-10-02496-f002:**
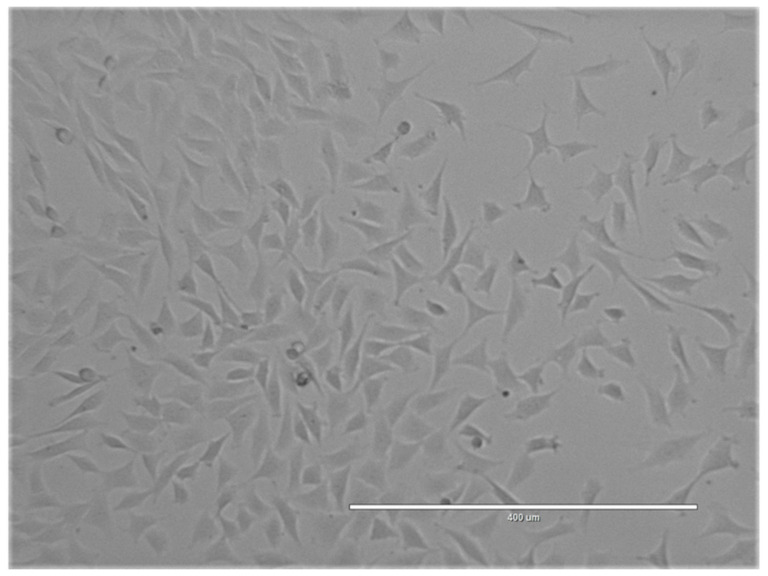
Expansion in culture of nasal chondrocytes derived from sheep nasal septum.

**Figure 3 cells-10-02496-f003:**
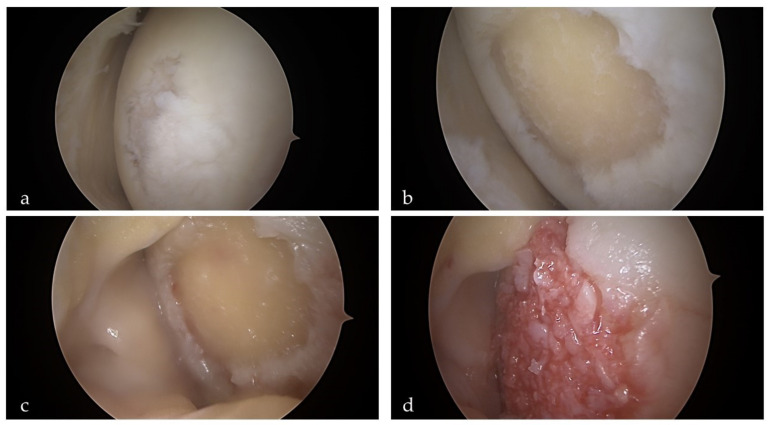
The single-step, full-arthroscopic autologous minced cartilage procedure for the treatment of knee cartilage defect. (**a**) Articular cartilage defect on the medial condyle of the left knee. (**b**) Defect after the debridement of all unstable cartilage with appropriate steep edges. During this step, chondral fragments are harvested for the preparation of the paste. (**c**) The arthroscopic fluid is drained from the knee, and the lesion is dried with a cotton swab. (**d**) Paste mixture containing autologous chondral fragments and platelet-rich plasma (PRP) is carefully placed into the defect. The last step involves the introduction of autologous thrombin serum. The combination of the fibrinogen contained in the paste and the thrombin applied creates a stable clot that holds the mixture in the lesion.

**Figure 4 cells-10-02496-f004:**
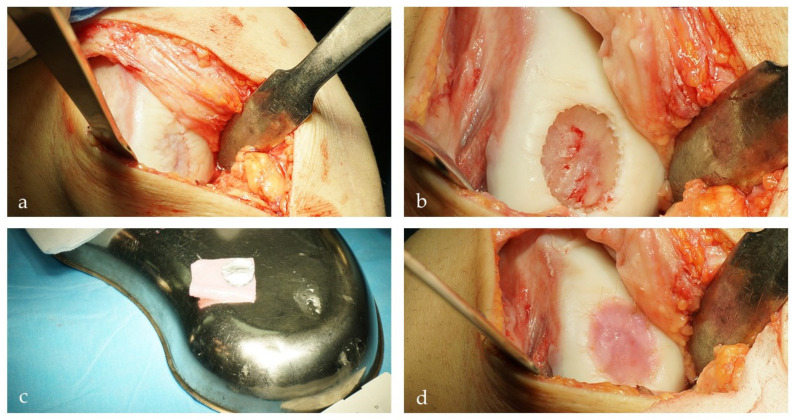
Transplantation of nasal-chondrocyte engineered construct for the treatment of the full-thickness cartilage defect in a 28-year-old athlete [19]. (**a**) Exposure of the full-thickness cartilage defect of the medial femoral condyle via mini-arthrotomy. (**b**) Debridement of the cartilage lesion to remove the damaged cartilage and establish adequate shouldering of the lesion. (**c**) Tissue-engineered cartilage cut to the right shape and ready for implantation. (**d**) Tissue-engineered cartilage inserted in the cartilage defect and secured by 5–0 monofilament absorbable sutures.

**Figure 5 cells-10-02496-f005:**
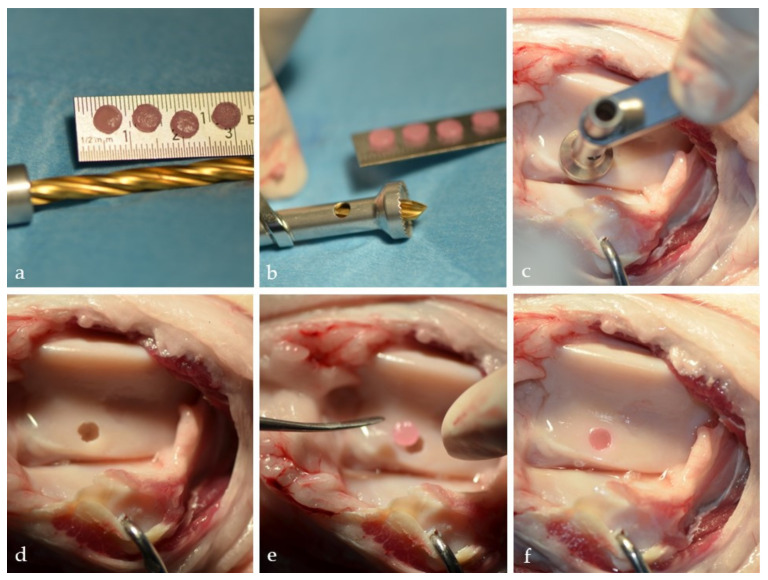
Wet lab mechanical stability testing of the iPSCs-based engineered cartilage construct. (**a**) Cartilage constructs engineered from iPSCs differentiated into chondrocytes. (**b**,**c**) Special drill was used to create defects on sheep trochlea. (**d**) Full-thickness cartilage defect on sheep trochlea. (**e**) Cartilage construct ready for press-fit transplantation. (**f**) Cartilage construct seated in the defect. Stable, no protrusion beyond the defect borders. (Cartilage constructs were grown and provided by Wa’el Kafienah, University of Bristol, U.K.).

**Table 1 cells-10-02496-t001:** Comparison of cell properties relevant to their application for cartilage treatment (number of pluses represent the scale where + means low, ++ moderate, +++ high, and ++++ very high).

Cells	AC	NC	CPs	BM-MSCs	AD-MSCs	Sy-MSCs	UB-MSCs	PB-MSCs	ESC	iPSCs
Source	autologous	autologous	autologous	autologous/allogenic	autologous	autologous	allogenic	autologous/allogenic	allogenic	autologous
Harvesting	difficult	moderate	difficult	difficult	moderate	difficult	easy	easy	difficult	easy
Availability	+	+	+	++	+++	+	+	+	++++	++++
Proliferation capacity	+	++	++	+++	+++	++++	++++	++++	++++	++++
Differentiation capacity	+	++	+++	+++	+	++++	+	+++	++++	++++
Safety issues	no	no	no	no	no	no	no	no	yes	yes
Ethical issues	no	no	no	no	no	no	no	no	yes	no

## Data Availability

Data sharing not applicable.

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
