# Peer review of "Cell Sources for Cartilage Repair—Biological and Clinical Perspective"

_cells, 2021, doi:10.3390/cells10092496_

Round 1

Reviewer 1 Report

Major:

  1. Disadvantages of BMSCs not outlined, by themselves they undergo chondrogenesis and then terminal chondrocyte differentiation to form calcified chondrocytes. In combination with chondrocytes (see the work of Daniel Saris) they actually perform as paracrine cells transiently providing the differentiating and survival signals needed to maintain surrounding chondrocytes within implanted tissue. Also it's interesting that Saris has pioneered the use of chondrons, i.e. chondrocytes still encapsulated in their original pericellular matrix, it's an innovation which needs to be explored in this review. - I have no connection with this research.
  2. Cartilage chondroprogenitors - tissue specific progenitors are also viable candidates for cell sources for cell therapies, and some space should be given over to their advantages/disadvantages. I have some connection with research in this area.

I think adding in these two areas will fully encapsulate what is happening in this research space and make for a better review.

Minor corrections:

line 17: ‘basic scientist’ - do clinicians describe themselves as ‘basic clinicians’? A scientist is a scientist and in this case i think you mean to say from the ‘biological scientists’ perspective’.

Line 18: ‘advantages and disadvantages’ rather than pro et contra - this will hopefully be read by people who would never have heard of latin never mind understanding it.

line 24: allows ‘smooth articulation of bones about each other’

line 26: ‘make’ for making.

line 27: remove Therefore.

line 36: Remove ‘Basic Science’ and replace with a less lazy heading.

line 58: ‘Replace ‘via’ with ‘By’.

line 77: remove ‘Articular’.

line 86: this sentence needs to be rewritten as it does not make sense.

line 88: ‘different studies have confirmed the latter findings’.

line 90: replace lost by lose.

line 154. Prof Brian Johnstone will be very disappointed to hear about this especially since it was he who devised this particular chondrogenic medium. Exp Cell Res. 1998 Jan 10;238(1):265-72. doi: 10.1006/excr.1997.3858. In vitro chondrogenesis of bone marrow-derived mesenchymal progenitor cells.

line 297: why are spheroids used?

line 377: with an equal differentiation potential  to that of . .

line 401: ‘study_patients’

line 439: ‘these types of embryonic-derived cell types’

line 479: abbr MR?

line 540: cell differentiation not cell faith!

Reference 78 looks incomplete - no literature or web citation just a name BIOCHIP - it references a clinical study.

Author Response

We have carefully reviewed the comments and have revised the manuscript accordingly. Our responses are given in a point-by-point manner below, except some grammar comments that are grouped together. Changes in the manuscript are marked with “track-changes” option in Word document.

Major corrections:

Comment 1:

Disadvantages of BMSCs not outlined, by themselves they undergo chondrogenesis and then terminal chondrocyte differentiation to form calcified chondrocytes. In combination with chondrocytes (see the work of Daniel Saris) they actually perform as paracrine cells transiently providing the differentiating and survival signals needed to maintain surrounding chondrocytes within implanted tissue. Also it's interesting that Saris has pioneered the use of chondrons, i.e. chondrocytes still encapsulated in their original pericellular matrix, it's an innovation which needs to be explored in this review. - I have no connection with this research.

Response to comment 1:

We added a text about BMSCs disadvantages and paracrine activity(lines 219-224).

„The major challenge of chondrogenic induction of BM-MSCs is controlling their differentiation because BM-MSCs tend to exhibit a hypertrophic phenotype leading to calcification [37,38]. The chondrogenic differentiation can be improved by coculture of BM-MSCs with chondrocytes. It has been suggested that coculture produces better cartilage matrix due to the BM-MSCs trophic role rather than their active chondrogenic differentiation [38–40]„

We also added a Saris's work with chondrons  (lines 91-97).

„Another advancement in the field of cartilage repair and regeneration is the use of chondrons – chondrocytes with pericellular matrix (PCM) [6]. The PCM is a thin layer of extracellular matrix (ECM) that surrounds chondrocytes and support their function [7]. The use of chondrons provides bioactive mechanical support for cells (with or without scaffold). This can be achieved through increased tissue surface by mincing cartilage tissue into smaller fragments that produce mitogenic signals and activate the migration of chondrocytes and ECM deposition [8].“

Comment 2:

Cartilage chondroprogenitors - tissue specific progenitors are also viable candidates for cell sources for cell therapies, and some space should be given over to their advantages/disadvantages. I have some connection with research in this area.

I think adding in these two areas will fully encapsulate what is happening in this research space and make for a better review.

Response to comment 2:

Cartilage chondroprogenitors definitely deserve a section in this review and we are sorry for this oversight.we included chondroprogenitors as Section 2.2.1. (line 164-178) and accordingly ajusted Figure 1and Table 1.

Minor corrections:

Comment 3:

line 17: ‘basic scientist’ - do clinicians describe themselves as ‘basic clinicians’? A scientist is a scientist and in this case i think you mean to say from the ‘biological scientists’ perspective’.

Response to comment 3:

We agree that our terminology is a little bit basic, and therefore we adopted your suggestion and changed the text and the title accordingly.

Changed to: “Cell sources for cartilage repair – biologicaland clinical perspective” (line 2)

 “In this paper we focus on the different cell types used in cartilage treatment, first from biological scientist perspective and then from clinician standpoint.“ (line 17)

Comment 4:

Line 18: ‘advantages and disadvantages’ rather than pro et contra - this will hopefully be read by people who would never have heard of latin never mind understanding it.

Response to comment 4:

Changed to: „We compare and analyze advantages and disadvantages of these cell types and offer potential outlook for future research and clinical application.“ (lines 18-19)

Comment 5:

line 24: allows ‘smooth articulation of bones about each other’

line 26: ‘make’ for making.

line 27: remove Therefore.

line 58: ‘Replace ‘via’ with ‘By’.

line 77: remove ‘Articular’.

line 88: ‘different studies have confirmed the latter findings’.

line 90: replace lost by lose.

line 401: ‘study_patients’

line 479: abbr MR?

line 540: cell differentiation not cell faith!

Response to comment 5:

All changes adopted as recommended.

Comment 6:

line 36: Remove ‘Basic Science’ and replace with a less lazy heading.

Response to comment 6:

Before: Basic science

Changed to: Biological perspective (line 46)

Comment 7:

line 86: this sentence needs to be rewritten as it does not make sense.

Response to comment 7:

The sentence is rewritten. (lines 112-114)

Before:“That raised a concern if extensive expansion of ACs in culture, limits their redifferentiation potential as they are expected to secrete neocartilage after implantation [9].”

Changed to:„If extensive expansion of ACs in culture limits their redifferentiation potential, that could affect neocartilage formation after ACs implantation [12].“

Comment 8:

line 154. Prof Brian Johnstone will be very disappointed to hear about this especially since it was he who devised this particular chondrogenic medium. Exp Cell Res. 1998 Jan 10;238(1):265-72. doi: 10.1006/excr.1997.3858. In vitro chondrogenesis of bone marrow-derived mesenchymal progenitor cells.

Response to comment 8:

The following sentence and reference to prof. Johnstone’s work is added to the text.

„Pellet culture is in vitromodel resembling mesenchymal precartilage condensations in embryo. It is favourable for cartilage formation because it mimicks cell-cell interactions. However, bioactive signals like dexamethasone  and TGF-β1 are important addition to growth medium [35].“

Comment 9:

line 297: why are spheroids used?

Response to comment 9:

The explanation about spheroids is added. (lines 450-453)

„A further modification of the ACI strategies is the use of chondrocytes cultured as small spheroids that are totally autologous without use of any foreign material, and are already in a higher developmental state then chodrocytes in suspension [84,85].“

Comment 10:

line 377: with an equal differentiation potential to that of . .

line 439: ‘these types of embryonic-derived cell types’

Response to comment 10:

These expressions are removed from the text because they are not written clearly and they are misleading.

Comment 11:

Reference 78 looks incomplete - no literature or web citation just a name BIOCHIP - it references a clinical study.

Response to comment 11:

Reference 78 is removed and clinical trial is listed in the text (line 545).

Thank you for the comments that made this manuscript more informative and valuable. We hope the revised version is now suitable for the Cells - Cell Therapies in Orthopaedics and we look forward to hearing from you in due course.

Sincerely,

Inga Urlic

Reviewer 2 Report

Urlic and Ivkovic provided a comprehensive review on the one of the three essential components in cartilage regeneration medicines, the cells. As they initially explain, the cells along with their tissue source and the stage of differentiation are of utmost importance for cell therapies beside scaffolds and signalling. The review is comprised of two main parts. The first part is providing the evidence on cartilage regeneration capabilities of mature, multipotent and pluripotent cells from basic and the second part from clinical point of view. The manuscript is generally well written and all the data can be easily taken in by the readers. There are some minor, however quite frequent grammar flaws and typos, in particular in the second part of the manuscript such as the lack of space between the words (lines 57, 204, 401, 453) or redundant space (line 178), redundant full stop (line 185), in vitro not written in italics (line 518), the abbreviation MSCs (lines 195, 477) or BM (line 482) not used consistently thorough out the manuscript, chondrocyte instead of chondrocytes (line 332), we instead of were (line 334), results instead of resulted (line 352) etc. I would advise the authors to thoroughly check the manuscript again by themselves or a third person with good grammar skills.

In addition to these minor issues, I would like to draw the authors attention to some additional issues listed below:

  1. In section 2.2. the authors state that the best sources of MSCs are bone marrow, adipose tissue and umbilical cord blood. As the word “best” is here very vaguely defined, I would suggest replacing this word with well-recognized or the most common as you have in section 2.2.1. In my opinion the best tissue source of MSCs for cartilage regeneration is still not known and as you mentioned several times there are pros and cons to every cell source.
  2. Synovium-derived MSCs have been identified by several basic studies for their propensity for chondrogenesis (continued work by Japanese group Sekiya and Muneta). Moreover, human synovium-derived MSCs have been the only MSCs so far identified to be morphogenic, i.e. forming a rudimental joint-like structure when modified to overexpress BMP7 (https://doi.org/10.1038/ncomms15040). This data should be also mentioned under section 2.2.3.
  3. In section 2.2.5. describing the PBMCs, the authors should first provide a few robust studies that these cells fulfil the criteria set by the ISCT in 2006 for MSCs. If there is not enough of the data, this section should be separated from MSCs section and stand alone.
  4. There is a legend or footnote missing in Table 1. The authors should clearly explain to the readers what they meant by +, ++, +++ and ++++. I would also suggest that both, the ethical and safety issues for ESC and iPSC should be written, as it is not clear in the current form what “yes” means for iPSCs. I suppose the authors meant the safety concerns associated with iPSCs as the ethical issues are ruled out as the authors explained in the text. I would suggest to write yes/yes and no/yes for ESC and iPSCs, respectively.
  5. The Figure 4 seems to be describing the study published in Lancet 2016 providing the evidence on the transplantation of nasal-chondrocyte constructs for the treatment of full-cartilage. This should be referenced in this Figure providing that the authors have the consent from the authors of these images.
  6. In section 3. Clinical perspective, the first sentence or two in each section are sometimes redundant as they are very similar to what has been written for each cell type in the section Basic science. Please uniform.
  7. There is a reference missing to the sentence: ‘In another case series, Sy-MSC were harvested, cultured in autologous human serum and arthroscopically implanted in 10 patients.’ (line 430). Please add.
  8. How can the authors claim that there are no ethical concerns associated with embryonic stem cells (line 439)? Please correct.
  9. Similarly as in section 2.2. I would suggest replacing the word ‘best’ with ‘optimal’ or ‘most favourable' in the last sentence in Conclusions.
  10. The authors should point out also the advantages and disadvantages of allogeneic versus autologous tissues as sources for cell therapies. Standards and requirements for the preparation of the allogeneic cell therapies and the presence of the concomitant degenerative disorders should be pointed out in particular for the usage of autologous cell therapies.

Author Response

We have carefully reviewed the comments and have revised the manuscript accordingly. Our responses are given in a point-by-point manner below, except some grammar comments that are grouped together. Changes in the manuscript are marked with “track-changes” option in Word document.

Comment 1:

Urlic and Ivkovic provided a comprehensive review on the one of the three essential components in cartilage regeneration medicines, the cells. As they initially explain, the cells along with their tissue source and the stage of differentiation are of utmost importance for cell therapies beside scaffolds and signalling. The review is comprised of two main parts. The first part is providing the evidence on cartilage regeneration capabilities of mature, multipotent and pluripotent cells from basic and the second part from clinical point of view. The manuscript is generally well written and all the data can be easily taken in by the readers. There are some minor, however quite frequent grammar flaws and typos, in particular in the second part of the manuscript such as the lack of space between the words (lines 57, 204, 401, 453) or redundant space (line 178), redundant full stop (line 185), in vitro not written in italics (line 518), the abbreviation MSCs (lines 195, 477) or BM (line 482) not used consistently thorough out the manuscript, chondrocyte instead of chondrocytes (line 332), we instead of were (line 334), results instead of resulted (line 352) etc. I would advise the authors to thoroughly check the manuscript again by themselves or a third person with good grammar skills.

Response to comment 1:

Thank you for pointing out grammar flaws and typos. We checked the manuscript thoroughly and made corrections of all listed and also other mistakes.

Comment 2:

In section 2.2. the authors state that the best sources of MSCs are bone marrow, adipose tissue and umbilical cord blood. As the word “best” is here very vaguely defined, I would suggest replacing this word with well-recognized or the most common as you have in section 2.2.1. In my opinion the best tissue source of MSCs for cartilage regeneration is still not known and as you mentioned several times there are pros and cons to every cell source.

Response to comment 2:

We agree that the word “best” is too bold so we made change as suggested. (Line 217)

Before: “the best sources of MSCs are bone marrow, adipose tissue and umbilical cord blood”

Changed to: “the well-recognized sources of MSCs are bone marrow, adipose tissue and umbilical cord blood”

Comment 3:

Synovium-derived MSCs have been identified by several basic studies for their propensity for chondrogenesis (continued work by Japanese group Sekiya and Muneta). Moreover, human synovium-derived MSCs have been the only MSCs so far identified to be morphogenic, i.e. forming a rudimental joint-like structure when modified to overexpress BMP7 (https://doi.org/10.1038/ncomms15040). This data should be also mentioned under section 2.2.3.

Response to comment 3:

Thank you for pointing out that we missed an important information about Sy-MSC. We included the data and appropiate references. (lines 310-322)

„Also, pellet cultures were significantly larger from Sy-MSC then BM-MSCs in patient-matched comparisons [57].“

„Combination of bone morphogenetic protein (BMP-2), TGF-β, and dexamethasone was optimal for induction of chondrogenesis Sy- MSCs [57]. In addition, the research of Roelofs et al. showed that Sy-MSCs carry an imprinted code for joint morphogenesis. They injected adult human Sy-MSCs overexpressing bone morphogenetic protein 7 (BMP-7) in the skeletal muscle and obsereved an ectopic formation of joint-like structure, providing the evidence of their morphogenetic properties [58].“

Comment 4:

In section 2.2.5. describing the PBMCs, the authors should first provide a few robust studies that these cells fulfil the criteria set by the ISCT in 2006 for MSCs. If there is not enough of the data, this section should be separated from MSCs section and stand alone.

Response to comment 4:

We apologize for the big mistake in interpretation of abbreviation PBMC as these are not all peripheral blood mononuclear cells but their subset called peripheral blood mesenchymal stem cells. The abbreviations are changed throughout the text to avoid confusion and the description of cell is explained in more detail as listed below. (lines 347-357)

„Circulating MSCs are also called peripheral blood-derived mesenchymal stem cells (PB-MSCs). The frequency of PB-MSCs in humans is very low, in the order of 1 in 108 peripheral blood mononuclear cells (MNCs) [63]. For comparison, the frequency of BM-MSCs in bone marrow is 1 in 104to 1 in 105bone marrow MNCs [64]. The BM-MSCs and PB-MSCs showed similar characteristics of cell proliferation and multi-differentiation potentials. Xu and Li compared the expression of PB-MSCs and BM-MSCs surface markers, and found that CD73 may be an important indicator to distinguish them. PB-MSCs are also plastic-adherent and have multi-differentiation potential fullfiling the criteria of  the International Society for Cellular Therapy[65].“

Comment 5:

There is a legend or footnote missing in Table 1. The authors should clearly explain to the readers what they meant by +, ++, +++ and ++++. I would also suggest that both, the ethical and safety issues for ESC and iPSC should be written, as it is not clear in the current form what “yes” means for iPSCs. I suppose the authors meant the safety concerns associated with iPSCs as the ethical issues are ruled out as the authors explained in the text. I would suggest to write yes/yes and no/yes for ESC and iPSCs, respectively.

Response to comment 5:

The explanation of signs is added to the table legend. The rows for safety and ethical issues are separated to avoid confusion. (lines 460-462)

Comment 6:

The Figure 4 seems to be describing the study published in Lancet 2016 providing the evidence on the transplantation of nasal-chondrocyte constructs for the treatment of full-cartilage. This should be referenced in this Figure providing that the authors have the consent from the authors of these images.

Response to comment 6:

The reference is added to the figure legend (line 550). The photos are taken by the co-author Alan Ivkovic during operation and we believe that the same photos have not been used elsewhere.

Comment 7:

In section 3. Clinical perspective, the first sentence or two in each section are sometimes redundant as they are very similar to what has been written for each cell type in the section Basic science. Please uniform.

Response to comment 7:

We removed the redundant sentences. However, some of the introductory section lines are left because they are connecting basic science to clinical procedures.

Comment 8:

There is a reference missing to the sentence: ‘In another case series, Sy-MSC were harvested, cultured in autologous human serum and arthroscopically implanted in 10 patients.’ (line 430). Please add.

Response to comment 8:

The missing reference is added as reference number 114. (Line 658)

Comment 9:

How can the authors claim that there are no ethical concerns associated with embryonic stem cells (line 439)? Please correct.

Response to comment 9:

We apologize for the misleading expression. We corrected the sentence. (Lines 655-656)

Before:“There are no ethical concerns associated with embryonic stem cells, and there is no donor site morbidity associated with other sources such as bone marrow of subcutaneous fat.”

Changed to: “Since there are no ethical concerns associated with their use, and there is no donor site morbidity associated with other sources such as bone marrow of subcutaneous fat.“

Comment 10:

Similarly as in section 2.2. I would suggest replacing the word ‘best’ with ‘optimal’ or ‘most favourable' in the last sentence in Conclusions.

Response to comment 10:

The sentences are corrected. (Lines 559-560)

Before:Taken all together, the best results are obtained with bone marrow-derived MSCs with moderate results in all categories.

Changed to:“Taken all together, the most favourable results are obtained with bone marrow-derived MSCs with moderate results in all categories.”

Comment 11:

The authors should point out also the advantages and disadvantages of allogeneic versus autologous tissues as sources for cell therapies. Standards and requirements for the preparation of the allogeneic cell therapies and the presence of the concomitant degenerative disorders should be pointed out in particular for the usage of autologous cell therapies.

Response to comment 11:

The paragraph about advantages and disadvantages of allogeneic versus autologous is added in Conclusion section. (Lines 850-855)

“Autologous transplants are golden standard when possible, but it is also important to emphasize some advantages of allogenic sources. Using of allogenic cells from different donors enables patients to undergo single surgical procedure. The tissue source is not as limited and the risk of infection is lower.  Allogenic transplants may carry the risk of immune rejection and disease transmission so the requirements to produce clinical-grade standardized of-the-shelf products are high.”

Thank you for the comments that made this manuscript more informative and valuable. We hope the revised version is now suitable for the Cells - Cell Therapies in Orthopaedics and we look forward to hearing from you in due course.

Sincerely,

Inga Urlic